# A social ecological approach to identify the barriers and facilitators to COVID-19 vaccination acceptance: A scoping review

Penny Lun[1]*, Jonathan Gao[1], Bernard Tang[1], Chou Chuen Yu[1], Khalid Abdul Jabbar[1], James Alvin Low[1,2,3], Pradeep Paul George[1,3,4,5]

**1** Geriatric Education and Research Institute Limited, Singapore, Singapore, **2** Geriatric Medicine, Khoo Teck Puat Hospital, Singapore, Singapore, **3** Lee Kong Chian School of Medicine, Nanyang Technological University, Singapore, Singapore, **4** Health Services and Outcomes Research, National Healthcare Group, Singapore, Singapore, **5** Faculty of Health Sciences, University of Adelaide, Adelaide, Australia

* lun.penny.sy@geri.com.sg

**Data Availability Statement:** All relevant data are within the paper and the supplemenotary files.

**Funding:** The author(s) received no specific funding for this work.

## Abstract

### Background

COVID-19 is an infectious disease caused by the SARS-CoV-2 virus that has caused substantial impact on population health, healthcare, and social and economic systems around the world. Several vaccines have been developed to control the pandemic with varying effectiveness and safety profiles. One of the biggest obstacles to implementing successful vaccination programmes is vaccine hesitancy stemming from concerns about effectiveness and safety. This review aims to identify the factors influencing COVID-19 vaccine hesitancy and acceptance and to organize the factors using the social ecological framework.

### Methods

We adopted the five-stage methodological framework developed by Arksey and O'Malley to guide this scoping review. Selection criteria was based on the PICo (Population, Phenomenon of interest and Context) framework. Factors associated with acceptance and hesitancy were grouped into the following: intrapersonal, interpersonal, institutional, community, and public policy factors using the social ecological framework.

### Results

Fifty-one studies fulfilled this review's inclusion criteria. Most studies were conducted in Europe and North America, followed by Asia and the Middle East. COVID-19 vaccine acceptance and hesitancy rates varied across countries. Some common demographic factors associated with hesitancy were younger age, being female, having lower than college education, and having a lower income level. Most of the barriers and facilitators to acceptance of the COVID-19 vaccines were intrapersonal factors, such as personal characteristics and preferences, concerns with COVID-19 vaccines, history/perception of general vaccination, and knowledge of COVID-19 and health. The remaining interpersonal, institution, community, and public policy factors were grouped into factors identified as barriers and facilitators.

**Competing interests:** The authors have declared
that no competing interests exist.

## Conclusion

Our review identified barriers and facilitators of vaccine acceptance and hesitancy and orga-
nised them using the social ecological framework. While some barriers and facilitators such
as vaccine safety are universal, differentiated barriers might exist for different target groups,
which need to be understood if they are to be addressed to maximize vaccine acceptance.

## Introduction

COVID-19 is an infectious disease caused by the SARS-CoV-2 virus that has resulted in signif-
icant impact on the health of the world's population, with over 500 million confirmed cases
and over 6.2 million deaths worldwide as of 3rd June 2022 [1]. Other consequences include
causing high levels of psychological distress as well as financial challenges for countries and
their healthcare systems [2, 3]. Hence, measures such as lockdowns, social distancing, and
travel restrictions were imposed by many countries in an attempt to reduce the spread of infec-
tions. While such measures can help [4], they are not long-term solutions.

Vaccinations can have an integral role in restoring normalcy [5]. While several viable can-
didates have been developed, approved, and distributed, uptake of these vaccines can be hin-
dered by vaccine hesitancy. A systematic review on COVID-19 vaccine acceptance rate found
varying rates among the public and health care professionals in different countries [6]. The
updated review also found variation of acceptance rates in different regions, with the Middle
East/North Africa, Europe and central Asia, and Western/Central Africa having more
COVID-19 vaccine hesitancy [7]. Common reasons cited for vaccine hesitancies are concerns
with the safety or effectiveness of the vaccines [8–13].

On the other hand, a study using large-scale retrospective data around the world found that
confidence in the importance of vaccines promoted vaccine uptake acceptance, whereas those
in minority religious groups were more likely to be hesitant [14]. An individual's political affil-
iation is also among the contributing factors to COVID-19 vaccine hesitancy [15, 16]. Clearly,
there is a need to consider the human factor to ensure widespread acceptance of COVID-19
vaccines [5]. Recommendations from the World Health Organization (WHO) centered
around the need to improve knowledge regarding vaccine safety and long-term effectiveness,
ensure accessibility, and employing targeted interventions to address the concerns of specific
populations [17].

To better understand the factors that posed as barriers or facilitated acceptance of COVID-
19 vaccines in the general public adult population, we conducted a scoping review using the
social ecological framework to segment the level of influences: Intrapersonal factors, interper-
sonal processes, institutional factors, community factors, and public policy [18, 19]. As vaccine
hesitancy is a complex issue that entails much more than one's attitude or behaviour [20, 21],
using the social ecological model could enhance understanding of external factors such as access
to the vaccines that involve wider system-related factors. The identified factors would also be
organized into barriers and facilitators to clarify if their influences on the acceptance or hesi-
tancy of COVID-19 vaccination are mutually exclusive or are more complexly intertwined.

While there have been other scoping reviews conducted on the determinants to vaccine
acceptance or hesitancy [11–13, 22], our review focused on studies conducted on the general
adult population that took place after the release of major COVID-19 vaccine clinical trial
results to account for possible shifting reasons for vaccine acceptance and hesitancy. This
review contributes by taking time sensitivity of the evolving barriers and facilitators into con-
sideration [23].

## Methods

We adopted the five-stage methodological framework developed by Arksey and O'Malley [24] with advancements to the framework proposed by Levac, Colquhoun and O'Brien [25] to guide this scoping review. In order to capture a variety of study types, the scoping review methodology was used. Quality assessments on the studies were not performed to allow for inclusion of more studies, as our purpose was to scope factors that could pose as barriers or facilitators to COVID-19 vaccine acceptance [24].

### The research question

The research question was refined through reviewing the literature on acceptance of the COVID-19 vaccines: What are the barriers and facilitators that affect the acceptance of COVID-19 vaccines among adults in the general public?

### Identifying relevant studies

The authors developed a search strategy with a medical librarian guided by the following key terms: COVID-19 or nCoV* or 2019nCoV or 19nCoV or COVID19* or COVID or SARS--COV-2 or SARSCOV-2 or SARSCOV2 or "Severe Acute Respiratory Syndrome Coronavirus 2" or "Severe Acute Respiratory Syndrome Corona Virus 2" [26] and Vaccination Refusal or Anti-Vaccination Movement or Mass Vaccination or Vaccination Coverage (hesitancy or acceptance or preference or rejection or anti-vaccination or attitude or barrier or facilitator or intention). The search strategy was adapted for PubMed, Medline, Embase, PsycInfo, and CINAHL and conducted on April 14, 2021. S1 Table shows the full search strategy in the respective databases. Reference lists of review papers found were also searched.

### Study selection

Selection criteria was based on the PICo (Population, Phenomenon of interest and Context) framework [27]. Table 1 presents details of the criteria. Time frame on the data collection (study) period rather than publication date was imposed for two reasons. Firstly, a recent scoping review reported on attitudes towards COVID-19 vaccination using the social-ecological model had included studies conducted before September 2020 [22]. Secondly, as people might

**Table 1. Selection criteria.**

|  | Inclusion criteria | Exclusion criteria |
|---|---|---|
| **Population** | • General Population<br>• Adults (15 and above) | • Population <15yrs old<br>• Specific segments of population (e.g., patients with certain diagnoses, health care professionals, specific professions) |
| **Phenomenon of Interest** | • Identified factors or barriers to COVID-19 vaccine acceptance/hesitancy<br>• Identified factors or facilitators to COVID-19 vaccine acceptance/hesitancy | • No factors identified or described on acceptance or hesitancy towards COVID-19 vaccines. |
| **Context** | • Regarding COVID-19 vaccines | • Regarding other vaccines/ outside of the COVID-19 pandemic context (e.g., influenza vaccine) |
| **Time frame** | • Data collection period from September 2020 onwards<br>• Data collection that started before September but continued in September 2020 and beyond | • Data collection that started and completed before September 2020 |
| **Filter: Study & publication types** | • Primary studies (All types including preprints of non-randomized interventions and RCTs)<br>• Publication in English | • Review papers<br>• Position statements<br>• Comments, Editorial, Opinions<br>• Grey literature |

change their minds with new information, we wanted to capture factors and perspectives that were more aligned with the actual COVID-19 vaccine situation after publication of the Pfizer and Moderna phase I/II trials data in the summer of 2020.

The title/abstract screening were conducted by three reviewers (BT, PL, JG) independently, with one main reviewer (BT) going through all titles and abstracts, while PL and JG shared the task as second reviewers. Any disagreements between two reviewers were discussed and resolved by the third reviewer (BT, PL, JG). During the full text review, each full text was again screened by two reviewers (combination of BT/PL, BT/JG, PL/JG) and disagreements were resolved by the third. As reported by Levac and colleagues [25], the screening process was iterative in order to refine the inclusion criteria. To calibrate our understanding during the screening process, disagreements were reviewed on a regular basis.

## Charting the data

The data extraction form was developed and pilot-tested before the start of the extraction phase. The extraction fields included publication details, study design and population, rates of COVID-19 vaccine acceptance/hesitancy, and reported factors influencing COVID-19 vaccine acceptance and hesitancy. Similar to studies that examined such factors using barriers and facilitators [28, 29], we defined factors associated with acceptance as facilitators and factors that deterred acceptance of the vaccination as barriers. These factors were then categorized using the social ecological framework [19] into intrapersonal factors, interpersonal processes, institutional/organizational factors, community factors, and public policy.

Intrapersonal factors include one's knowledge, attitudes, behavior, self-concepts, and skills, whereas interpersonal factors are the influences from formal and informal social networks [19]. Institutional factors describe "social institutions and organization characteristics, and formal (and informal) rules and regulations for operations" [19] that impact people's acceptance of the COVID-19 vaccines and vaccination. Community factors refer to relationships among organizations and groups with boundaries defined by geographical and political terms, whereas public policy refers to policies or programs that promote or deter certain behaviors [19]. To better conceptualize each level in the social ecological model, sub-categories defined in other vaccine studies using the framework were adapted to guide allocation of the extracted factors [30, 31]. To ensure objectivity, data from each included article was extracted by one reviewer independently (BT, JG, or PL) and 20% of the extracted data were cross-checked by a second reviewer (BT, JG, or PL). The screening and data extraction were conducted in Covidence systematic review software [32]. S3 and S4 Tables show the data extracted and the excluded full text with reasons respectively.

## Collation, summarizing, and reporting the results

The results were collated and summarized descriptively by the three reviewers (BT, JG, PL). The factors and themes (when necessary) identified were classified under barriers or facilitators in the social ecological framework. These were shared and discussed with the remaining research team before finalization. Preferred Reporting Items for Systematic Reviews and Meta-Analyses Extension for Scoping Review (PRISMAScR) was used to report this scoping review (see S5 Table) [33].

## Results

A total of 1066 unique citations were found using our search strategy. After the title/abstract screening, 233 citations were included for full text screening, of which 51 articles fulfilled the inclusion criteria for synthesis. Most of these articles were published in 2021(n = 46, 90%),

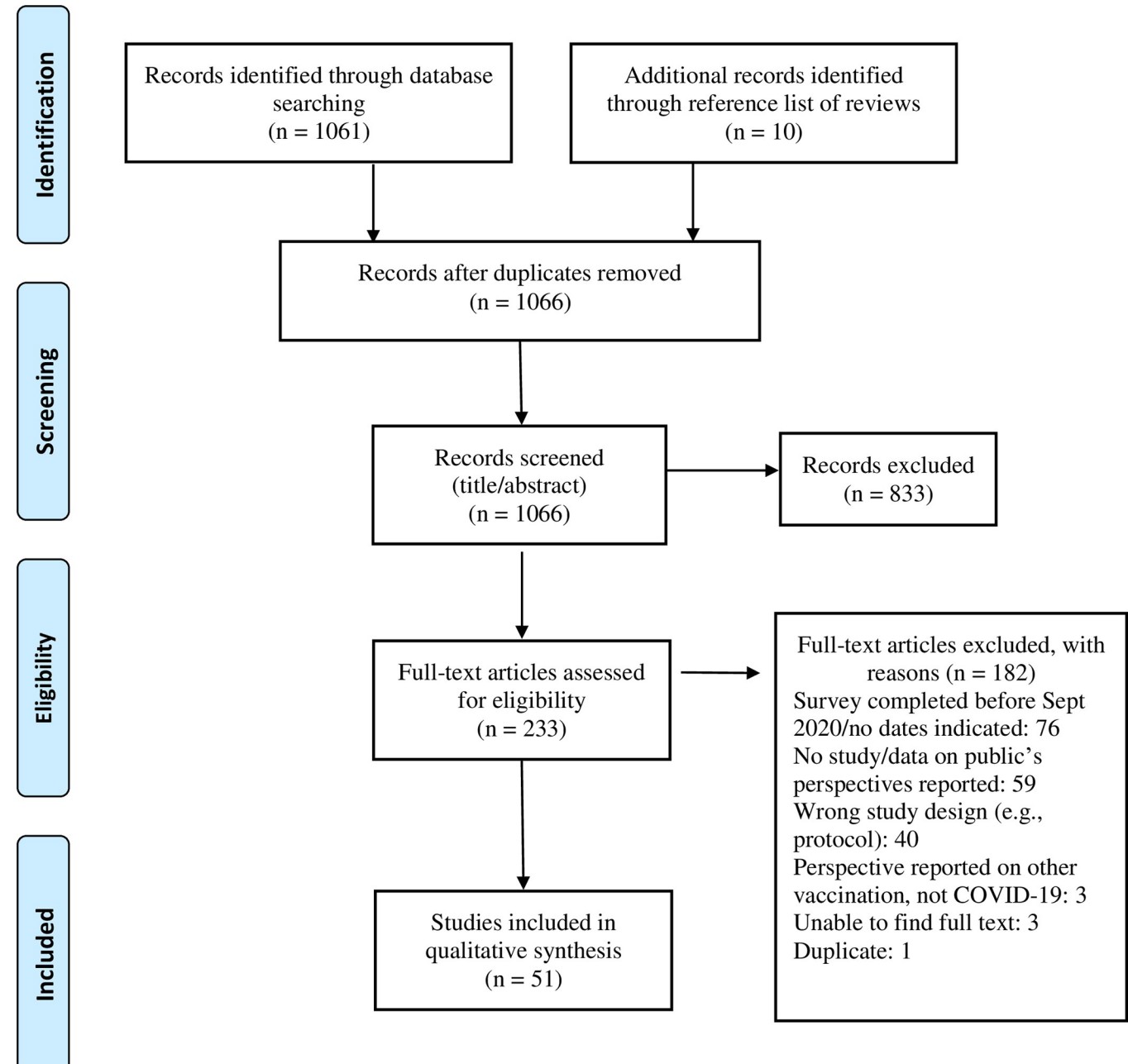

**Fig 1. Preferred reporting items for systematic reviews and meta-analyses flow diagram of study selection process.**

whereas the rest were published in 2020. Fig 1 shows the PRISMA flow diagram of the screening process for the review [34].

## Study characteristics

A Majority of the studies were cross-sectional studies (n = 32, 63%), with data collected via surveys (n = 45, 88%). Most of the studies had sample sizes over 1000 (n = 32, 63%) and included participants from Europe (n = 15, 29%), North America (n = 15, 29%), Asia (n = 9, 18%), and the Middle East (n = 5, 10%). In addition, a majority of the studies included adults from 18

**Table 2. Study characteristics.**

| Study Characteristics | Category | Numbers of studies n (%) |
|---|---|---|
| **Study design** | Cross-sectional | 32 (63) |
| | Experiment (e.g. RCT) | 5 (10) |
| | Longitudinal/Cohort | 5 (10) |
| | Qualitative | 1 (2) |
| | Others (e.g.social media analysis, multi-methods, discrete choice experiment) | 8 (16) |
| **Data collection method** | Survey (online, phone) | 45 (88) |
| | Interview (phone) | 1 (2) |
| | Focus Group | 1 (2) |
| | Use of public data (social media) | 4 (8) |
| **Sample size** | <1000 | 11 (21) |
| | 1000–4999 | 21 (41) |
| | 5000–10000 | 5 (10) |
| | >10000 | 6 (12) |
| | Multiple samples* | 4 (8) |
| | Unknown** | 4 (8) |
| **Study region** | Europe (Italy, Spain, UK, Portugal, Germany, Greece, Slovenia) | 15 (29) |
| | North America (USA, Canada) | 15 (29) |
| | Asia (China, India, Japan, Hong Kong) | 9 (18) |
| | Middle East (Jordan, Saudi Arabia, Qatar, Israel, Kuwait) | 5 (10) |
| | Others (Brazil, Congo, Russia) | 3 (6) |
| | Multi-countries | 4 (8) |

*Studies with more than one sample due to >1 time-point measurements or >1 study embedded
**Social media data without population sample size

years and above, while 6 studies included participants from 15–16 years [35–40]. One study was conducted solely on older adults [41]. Nineteen studies reported the mean age of their overall study participants, which were mostly in the 40s. Table 2 shows study characteristics (S2 Table provides study characteristics details).

## Acceptance/Hesitancy rate

Thirty-seven of the 51 studies reported COVID-19 vaccine acceptance and/or hesitancy rates. Most of these studies assessed one's willingness or intentions to take the COVID-19 vaccines when available, using Likert-type scales, instead of validated scales such as the Vaccination Attitudes Examination (VAX) Scale [42, 43]. One study measured trust in the vaccines [44]. For the purpose of our study, the hesitancy rates included those who: (a) reported unwilling-ness to accept the vaccination, (b) were strongly hesitant, or (c) reported being very unlikely to accept COVID-19 vaccination.

Global acceptance rates ranged from 13.1–91%, and hesitancy rates ranged from 7–86.6%. Studies from Spain (77.6%), Germany (78.2%), China (81.3–88.6%), and India (83.6%) reported acceptance rates over 70%. Some studies in Italy [45], UK [35, 40, 46], and USA [47] also reported acceptance rates over 70%. On the other hand, studies from Hong Kong [48] and several middle Eastern countries [36] reported acceptance rates of below 30%. Acceptance and/or hesitancy rates could not be determined in 13 studies, as they were experimental studies

**Table 3. COVID-19 vaccine acceptance and hesitancy rates by country.**

| Country (n) | Acceptance Rate (%) | Hesitancy Rate (%) |
|---|---|---|
| Italy (3)* | 53.7–91 | 46.2–59.1 |
| Spain (1) | 77.6 | 22.4 |
| Portugal (1) | 35.3 | 65 |
| Slovenia (1) | 59 | NA |
| Germany (1) | 78.2 (including those already vaccinated) | 21.8 |
| UK (6)* | 54–82 | 7–46 |
| USA, UK (1)** | 48.6(UK) <br> 39.8(US) | 42.4 (UK) <br> 42.7 (US) |
| USA (7)* | 39.4–81.5 | 18.5–60.6 |
| China (2)* | 81.3–88.6 | 11.4–18.7 |
| Japan (2)* | 62.1–65.7 | 34.3–37.9 |
| Hong Kong (1) | 13.1 (Soonest) | 86.6 |
| India (1) | 83.6 | 16.4 |
| Jordan, Kuwait, Saudi Arabia, others (1) | Overall:29.4 Jordan-28.4, Kuwait-23.6, Saudi Arabia-31.8, Others-41.5 | Jordan-71.6, Kuwait-76.4, Saudi Arabia-68.2, Others-58.5 |
| Jordan (1) | 36.8 | 63.2 |
| Saudi Arabia (1) | 48.4 (If free) | 51.6 |
| Qatar (1) | 60.5 | 39.5 |
| Kuwait (1) | 53.1 | 46.9 |
| Brazil (1) | 88 | 12 |
| Russia (1) | 41.7 <br> 63.2 (if proven safe and effective) | NA |
| Congo (1) | 55.9 | N/A |
| Malta and International (1) | 51 | 48.2 |

*Countries with more than 1 study display only the range of the proportions of acceptance and hesitancy (if available), the figures are not meant to add up.

**Results of acceptance following RCT on exposure to misinformation

or social media studies that measured positive and negative sentiments towards the vaccines, or were studies that presented only stratified results (*e.g.*, gender, ethnic). Table 3 summarizes the acceptance and hesitancy rates by country.

## Barriers and facilitators using social ecological framework

**Socio-demographic determinants.** In general, most studies found that lower age, being female, having lower than college education, having a lower income level, and having or living with school age children were commonly cited determinants of vaccine hesitancy and rejection. A minority of the studies found opposite trends. For example, two studies found that males were more hesitant than females [49, 50], whereas one study found that higher education was a barrier [48] and another where having the least education was a facilitator [51]. With regards to age, a positive trend of COVID-19 vaccine acceptance among older adults above 60 to 70 were reported in several studies [9, 10, 38, 48, 52–54]. However, one study conducted in Qatar found that being older than 65 was a significant factor in vaccine hesitancy, compared to the younger participants surveyed [43]. Interestingly, a discrete choice experiment study found that those aged 55 and above were significantly more likely to reject a vaccine with 50% protection, while being significantly more likely to accept a vaccine with 90% protection, when compared to the younger participants [40].

In addition, a few studies that identified ethnic minority as a determinant of vaccine hesitancy were conducted in the UK and USA [10, 35, 46, 47, 52, 55–57]. Although few studies measured political leanings, their results were consistent, where those identifying with conservative ideology tended to be more hesitant [44, 58], compared to those identifying with liberal/democratic political ideology [44, 52, 58, 59]. Having comorbidities could either be a barrier, when one has compromised immunity or conditions in which the vaccine was not recommended [44, 60], or a facilitator [10, 36, 41, 48–50, 53, 54, 61, 62] where one identifies as being vulnerable to the severe effects of COVID-19. Likewise, perceived fair or good health could be a barrier [9, 52] or perceived reasonable health as a facilitator [61] to one's willingness to be vaccinated. Current smokers [65, 72], and those who have psychological distress, were less likely to accept the vaccine [56]. Table 4 shows a full list of the social demographic factors identified.

**Intrapersonal factors.** As there were many identified factors in this category, they were further categorized into broader themes, namely: Individual characteristics and preferences, concerns with COVID-19 vaccines, history/experience of vaccination, and knowledge/perception of COVID-19 and health-related information. Fig 2 summarizes factors/themes identified in the social ecological framework.

*Individual characteristics and preferences*. Those who have had COVID-19 infections and believed that they have acquired immunity [55, 60], and those who preferred to acquire immunity naturally, were less likely to accept the vaccination [37, 38, 42–44, 50, 65, 72]. Those who experienced negative emotions (felt agitated, sad, or anxious) or adhered less with health measures and guidelines, such as wearing a mask and safe distancing, or not having been quarantined, were also less likely to accept the vaccine [42, 46, 49, 61]. Having a strong perception of one's autonomy [71] and not believing that being vaccinated was one's civic duty as a citizen [47] were barriers to accepting the vaccines as well. On the other hand, the belief that getting vaccinated protected others or was one's social responsibility was associated with acceptance towards COVID-19 vaccination [29, 35, 47, 50–53, 70, 73, 74]. The desire to return to normalcy also facilitated acceptance of the vaccines [35, 54, 74, 75]. Lastly, one study conducted in Saudi Arabia found that those believing in mandatory vaccinations were more likely to accept the COVID-19 vaccination [69].

*Concerns with COVID-19 vaccines*. Our findings identified that major barriers to acceptance of COVID-19 vaccines were doubts or mistrusts on their effectiveness and benefits [9, 29, 38, 42–44, 50, 51, 54, 55, 60, 62, 65, 66, 72, 74, 76, 77], including not having enough information or evidence [38, 65, 66, 74]. Perceived risk of the vaccines from their potential side effects or unforeseen longer-term effects were also identified as barriers in many studies [9, 10, 29, 35, 38, 42, 48–50, 54, 55, 58, 60, 62, 65, 72, 73, 76–78]. Other perceived barriers included concerns over the following: Requiring more than one dose or booster doses, convenience (*e.g.*, timing of vaccination appointment), not being able to receive proof of vaccination, or the need to submit personal information etc. [29, 40, 49, 52, 78]. Conversely, perceiving higher benefits than risks, such as high vaccine efficacy, facilitated acceptance of the vaccines [29, 35, 38, 41, 48, 50, 52–54, 65, 67, 68, 71, 73, 78]. Accessibility influences acceptance/hesitancy as well, with studies suggesting that ease of access relates to higher acceptance [51, 53, 59, 68].

*History/experience of vaccination*. Those who had a history of rejecting vaccines, including flu vaccines and those who mistrusted vaccines in general, were less likely to accept the COVID-19 vaccines [9, 35, 38, 42, 44, 50, 54, 55, 60, 61, 66, 69, 72, 76, 78], whilst those who had a history of being vaccinated were more likely to accept them [9, 37, 38, 43, 48, 52, 59, 61, 65, 69, 73, 74]. Factors such as having had adverse reactions or bad experiences with previous vaccinations [9, 37, 38, 74], and being fearful of needles or injection [38, 50, 54, 55], contributed to COVID-19 vaccine hesitancy.

**Table 4. Socio-demographic factors.**

| Factors | Associated with hesitancy | Associated with acceptance |
|---|---|---|
| **Age** | Younger age [9, 35, 42, 46, 48–50, 53–57, 63, 64]. | Younger age [47, 65].<br>Higher age (but not above 60) [10, 29, 37, 39, 50, 51, 61]. |
| | Age 55+(when vaccine offers 50% protection) [40].<br>Age 55–64 [65].<br>Age 65 and above [43]. | Age 55+ (when vaccine offers 90% protection [40].<br>Age 60 and above [38, 52].<br>Age 65 and above [9, 10, 48, 53, 54]. |
| **Gender** | Being female [9, 10, 35, 38, 42, 43, 46, 47, 53, 55–57, 61, 63, 66]. | Being female [45, 50]. |
| | Being male [49, 50]. | Being male [9, 36–38, 44, 48, 51–54, 59, 61, 63, 65, 67–69]. |
| **Race** | Ethnic minority [35, 47, 56].<br>Black and mixed ethnicities [46].<br>Non-Hispanic Black (Black) [55]. African American [52].<br>Black [10, 35].<br>Non-white [57].<br>Non-Hispanic White [49].<br>Older female Arabs [63]. | Other race (not White/Hispanic/Black) [10].<br>Non-Hispanic Black [44].<br>Non-Black [59].<br>Asians [51]. |
| **Education** | Lower education(lower than a degree) [10, 42, 46, 47, 49, 53, 55, 56, 61, 63]. | Those with the least education [51]. |
| | Higher education [48].<br>Diploma degree [66]. | Higher education (at least a degree) [35, 36, 49, 50–53, 57, 61, 62]. |
| **Income** | Low Income [42, 46, 53, 55, 64].<br>Lost income during pandemic [61]. | Middle to High income [10, 52, 53, 57, 62, 69].<br>Low income [67]. |
| **Socio-economic status** | NA | Higher socio-economic status [29]. |
| **Home owner** | Not a home owner [46]. | Home owner [46]. |
| **Religion** | Religious reasons (not specified) [50, 54]. | Jewish [63].<br>Jewish, Muslims, atheist, and others [70].<br>Catholic, Protestant, Adventist, Pentecostal, Revival [62]. |
| **Political Leanings** | Conservative [44, 58]. | Liberal/Democratic political Ideology [44, 52, 58, 59]. |
| **Occupation** | Workers [61].<br>Not employed full time, not retired, a change in working [46].<br>Health care workers [62].<br>Non-medical staff, students [50]. Retiree [65].<br>Key workers [42].<br>First responders, construction, maintenance and landscape, homemakers, housekeeping, cleaning and janitorial, retail and food service workers [51]. | Working full-time [46].<br>Working in close contact with public [9].<br>Studying/working in healthcare [37, 50, 51, 60, 65, 66].<br>Essential worker [49].<br>Office/professional/technical workers/educators [39, 51].<br>Unemployed [51, 70].<br>Retired [46, 48, 51, 61].<br>Housewives [48].<br>Students [51, 61, 66]. |
| **Marital Status** | Single/not married [48, 53].<br>Married [46, 66]. | Single/Widowed/Divorced/not married [41, 46, 66].<br>Married [53]. |
| **Residency status** | Being a native [43]. | Being a Foreigner [65]. |
| **living area** | Living in non-metropolitan areas [55]. Central areas [54]. | Living in metropolitan [52].<br>Living in rural areas [54]. |
| | Regions in a country [50, 62, 65]. | Regions in a country [50, 62, 65, 69]. |
| **Have Children** | have school age children/children below 18/living with children [42, 47, 48, 61, 66]. | Not having a child at school [46]. |
| **Functioning** | Lower cognitive scores [56]. | Having a disability [51]. |
| **Comorbidity/chronic illness** | Less likely to have chronic disease/at high risks for COVID-19 [49, 52, 53, 55, 56].<br>Have chronic conditions/compromised immune that the vaccine was not recommended [44, 60]. | With chronic condition, comorbidities [10, 36, 41, 48–50, 53, 54, 61, 62]. |

(*Continued*)

**Table 4.** (Continued)

| Factors | Associated with hesitancy | Associated with acceptance |
|---|---|---|
| **Health/Mental Health status** | Perceived fair or good health [9, 52].<br>Have psychological distress [56]<br>Having fear [9]. | Perceived health status as reasonable compared to good or very good [61].<br>To achieve peace of mind [71].<br>Lower self-rated overall health [65].<br>Under/over-weight [65].<br>Does not consume vitamin C [66]. |
| **Relating to COVID-19** | Already had COVID-19 and hence immunity [9, 60]. | Previous or current infection [69]. |
| **Smoker** | Current Smoker [65, 72]. | Former smoker [65]. |

*Knowledge/perception on COVID-19 and health-related information.* Those who perceived a high risk of contracting COVID-19, and/or possible severe consequences from COVID-19, were more likely to accept the vaccination [41, 44, 46, 47, 51, 52, 54, 59, 61, 65, 66, 68, 69, 73, 78], whereas those who perceived a low or non-existing risk of infection, or believed that they were less likely to develop complications, were less likely to accept the COVID-19 vaccination [52, 54, 55, 57, 61, 65, 77]. The latter group included those who believed that the COVID-19 symptoms were mild or had been exaggerated. In addition, belief in conspiracy theories (*e.g.*, COVID-19 is hoax, vaccines are used to control or kill people) was also found to be a barrier to COVID-19 vaccination acceptance [36, 37, 46, 57, 60, 62, 66]. Table 5 shows full list of the intrapersonal factors.

**Interpersonal factors.** Influence from those who were vaccinated, influence from those infected with COVID-19, and normative influences from family and friends were three factors found that impact one's perception and attitude toward COVID-19 vaccination acceptance. Knowing someone with a serious vaccine reaction [52] or not knowing anyone close who was affected by COVID-19 [64] were barriers to one's acceptance, whilst knowing someone who was infected [66], or knowing people who had been or intended to get vaccinated [29, 48, 54],

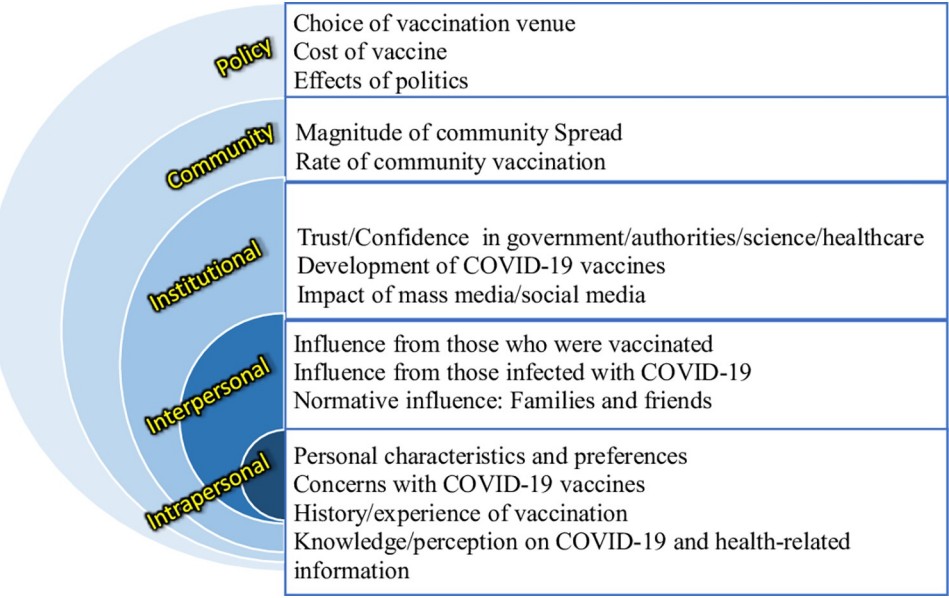

**Fig 2. Summary of factors/themes in the social ecological framework.**

**Table 5. Intrapersonal factors.**

| Themes | Factors | Barriers | Facilitator |
|---|---|---|---|
| **Individual Characteristics/ Preferences** | Naturalness bias | Naturalness bias/prefers natural immune [37, 38, 42–44, 50, 65, 72].<br>Already been immune as a result of being infected [55, 60]. | Lower naturalness bias [59, 65, 67].<br>Disagree that one does not have to vaccinate because of natural immunity [45]. |
| | Perception of vaccination as a social responsibility or civic duty | Disagree that being vaccinated-civic duty [47]. | Believe getting vaccinated is social responsibility/protect others/good prevention [29, 35, 47, 50–53, 70, 73, 74]. |
| | Willing to take risk | Not willing to be among the first to take the vaccine [38] | Willing to take risk [68]. |
| | Acquiring resources mindset | N.A. | Acquiring resources mindset [68]. |
| | Preventive behaviours | Negative emotions towards/less adherence to health measures or guidelines [42, 46, 61]. | Engage in preventive behavior [52, 66]. |
| | | Have not been quarantined [49]. | Have been tested for COVID-19 [62]. |
| | | Other preventive measures were enough [54, 55]. | No longer need preventive measures after vaccination [54]. |
| | Autonomy Vs Discussion/ Mandatory vaccination | Stronger Perception of autonomy [71]. | Likely to discuss COVID-19 with healthcare provider [52].<br>Support mandatory vaccination [69]. |
| | Willing for others around them to be vaccinated | N.A. | Willing for others around them to be vaccinated [38]. |
| | | NA | Desire to return to normalcy (e.g. travel) [35, 54, 74, 75]. |
| **Concerns with COVID-19 vaccines** | Perceived accessibility of vaccines | Perceived barriers (e.g. year booster shots, need to submit personal inform to get vaccination, vaccination convenience, vaccine availability) [29, 40, 52, 78].<br>Not providing proof of vaccination [49]. | Able to easily access vaccine [51, 53, 68].<br>Perceived capacity to get to vaccination site [71]. |
| | Perceived benefits of vaccination | Doubt/mistrust on vaccines-effectiveness/ risk-benefits/ not reliable due to being new/wait and see [9, 29, 38, 42–44, 50, 51, 54, 55, 60, 62, 65, 66, 72, 74, 76, 77].<br>Lack of vaccine information/want more evidence [38, 65, 66, 74]. | Perceived benefits of vaccination [29, 35, 38, 41, 48, 50, 52–54, 65, 67, 68, 71, 73, 78].<br>Sufficient data is available about the vaccine [29]. |
| | | Lower than 50% efficacy [40]. | Having high vaccine efficacy of at least 71% or 90% [40, 48, 78]. |
| | Perceived risk of vaccination | Concerns with side effects/serious adverse reactions/ unforeseen future effects/long term impact on health [9, 10, 29, 35, 38, 41, 42, 48, 50, 54, 55, 58, 60, 62, 65, 72, 73, 76, 77, 78]. | Low perceived risk of vaccination [29, 45, 51–53, 58, 65, 67]. |
| | | Vaccine could give me COVID [44, 55, 74, 79]. | NA |
| | Type of Vaccine | NA | Inactivated vaccines [80]. Innovative technology used in developing vaccine [47]. Open to novel vaccine [50]. |
| **History/experience of vaccination** | History of vaccination | Not vaccinated against flu /anti vax attitude/mistrust in vaccines/history of rejecting vaccines [9, 35, 38, 42, 44, 50, 54, 55, 60, 61, 66, 69, 72, 76, 78].<br>Previous vaccine adverse reaction/bad experiences [9, 37, 38, 74]. | Previous vaccination [9, 37, 38, 43, 48, 52, 59, 61, 65, 69, 73, 74].<br>Positive general attitude to vaccines [57]. |
| | | Fear of needles and injections [38, 50, 54, 55]. | NA |
| **Knowledge/perception on COVID-19 and health-related information** | Perceived risk of COVID-19 | Perceiving a low/ non-existing risk of infection/ developing complications/effects are mild, exaggerated [52, 54, 55, 57, 61, 65, 77].<br>Perceived high risk of infection [48]. | Perceived moderate to high risk of COVID-19 infection/severity of infection [41, 44, 46, 47, 51, 52, 54, 59, 61, 65, 66, 68, 69, 73, 78].<br>Believe COVID-19 exists [62]. |
| | Ability to understand information | Deficit in medical and epidemiologic literacy [76]. | Able to understand COVID-19 relevant information [61, 43]. |
| | Lack of knowledge/ uninformed | Poor knowledge on COVID-19 [9, 42]. | N.A. |

*(Continued)*

**Table 5.** (Continued)

| Themes | Factors | Barriers | Facilitator |
|--------|---------|----------|-------------|
| | Facts Vs fake news | Exposure to misinformation [70]. | Exposure to facts [70]. |
| | Conspiracy theory | Believe in conspiracy theory (e.g. COVID-19 is hoax, vaccines used to control or kill people) [36, 37, 46, 57, 60, 62, 66]. | N.A. |

facilitated acceptance. In addition, trusting information or valuing opinions of family, friends, and acquaintances could either be a barrier [37] or a facilitator [38, 71], depending on their own views on hesitancy or acceptance. For example, one study found that the opinion of family and friends was significantly correlated with participants' willingness to take a COVID-19 vaccine [38].

**Institutional/Organizational factors.** Four institutional/organizational factors were found to influence one's acceptance of the vaccine: (a) Trust/confidence in governments/ authorities, (b) trust/confidence in science, and/or healthcare/healthcare personnel, (c) development of COVID-19 vaccines, and (d) impact of social media vs. legacy media (*e.g.*, newspapers, radio, television). Studies that measured social media responses were included in this category.

Low trust or mistrust toward the government and health authorities, such as WHO and the Centres for Disease Control and Prevention (CDC), could pose as a barrier to one's acceptance of the COVID-19 vaccination [37, 40, 43, 44, 47, 52, 55, 61, 63, 66, 74], whereas high trust and confidence in the government and health authorities promoted acceptance of the vaccines [10, 37, 44, 48, 52]. Similarly, low trust or mistrust of the pharmaceutical industry and healthcare providers would make one less likely to accept the COVID-19 vaccines [44, 46, 47, 52, 61, 66, 76, 77], including those who were concerned with profiteering by the pharmaceutical companies [29, 42, 72]. A lack of recommendation by a doctor has been cited as a reason for not intending to take the COVID-19 vaccine [55], whereas having a recommendation from a doctor influenced vaccine acceptance [67, 78].

In addition, the speedy development of the vaccines [29, 55, 60, 76, 77], the manufacturing country [48, 77, 81], and the perceived lack of information or clear data (*e.g.*, inconsistent/ contradictory/delays and trial pauses) contributed to people's hesitancy [9, 60, 61, 79]. While trusting alternative sources of information, and exposure to coverage on WhatsApp, blogs, and social media were more likely to increase one's hesitancy toward the vaccines [37, 40, 57], one study found that frequent exposure to positive social media messages facilitated one's acceptance [48]. As expected, those who trusted official sources of information and legacy media were less likely to be hesitant toward the vaccines [37, 44, 57, 75, 79]. A point to note is that those who thought that the media over-reported on the side effects of the vaccines facilitated their acceptance of the vaccines [41].

**Community factor and public policy.** Low magnitude of community spread (*e.g.*, number of confirmed or suspected cases in the county) could pose as a barrier to vaccine acceptance [78]. In addition, rate of community vaccination could either result in low perceived risk, which would be a barrier [50] or a facilitator, if that nudged one into getting vaccinated [67]. Meanwhile, the choice of the vaccination venue could either be a barrier [40, 49] or a facilitator [40] to one's willingness to be vaccinated, depending on if their preferences were met. Cost of the vaccine [39, 78] or associated cost with the vaccines [55] could also be a barrier to some, especially for those without health insurance [55].

The effects of politics have spilled over to influence people on their hesitancy or acceptance of the vaccines. Those who believed that political pressure had influenced the development speed of the vaccine were more likely to be hesitant toward accepting the vaccines, and

**Table 6. Other factors (interpersonal, institutional, community, public policy).**

| Interpersonal | | |
|---|---|---|
| **Factors** | **Barriers** | **Facilitators** |
| **Influence from those who were vaccinated** | Knowing someone with a serious vaccine reaction [52]. | People around me (including role models) have been vaccinated or intend to get vaccinated [29, 48, 54]. |
| **Influence from those infected with COVID-19** | Not having anyone close who had been affected by COVID-19 [64]. | Knowing someone infected with COVID-19 [66]. |
| **Normative influences: Families and friends** | Trusting information from friends and acquaintances (non-Healthcare profession) [37]. | Valuing opinions of family/friends or those in this group who share the same views/beliefs [38, 71]. |
| | N.A. | Descriptive norms (i.e. believing people similar or important to you would get COVID-19 vaccine)/ Social norms on COVID-19 prevention [44, 73]. |
| | NA | Frequency of socializing prior to the pandemic [63]. |
| **Institutional Factors** | | |
| **Factors** | **Barriers** | **Facilitators** |
| **Trust/ confidence in governments/authorities** | Mistrust/low trust/dissatisfactions towards government/authority or their handling of the pandemic situation [37, 40, 43, 44, 52, 55, 61, 63, 66, 74]. | High trust/confidence/satisfactions towards government and authorities (e.g. WHO, CDC) [10, 37, 44, 48, 52]. |
| **Trust/Confidence in science and healthcare** | Mistrust/low trust /concerns towards pharmaceutical industry, healthcare providers, science [44, 46, 47, 52, 61, 66, 76, 77]. Concerns with profiteering by pharmaceutical companies [29, 42, 72]. | Having high confidence and trust in the healthcare system (effectiveness and positive experiences) [37, 38, 44, 46, 47, 57, 61, 67]. |
| | No recommendations from doctors/health authorities [50, 55]. | Recommended by trusted doctors [67, 78]. |
| **Development of COVID-19 Vaccines** | Development speed was rushed [29, 55, 60, 76, 77]. | N.A. |
| | Manufacturer/Country [48, 77, 81]. | Manufacturer and type [48, 80]. |
| | Lack of information or clear data (e.g. inconsistent/ contradictory/ delays and trial pauses) that concerned people [9, 60, 61, 79]. | If vaccine is demonstrated/proven to be safe [35, 67, 79]. |
| **Impact of social media Vs legacy media** | Coverage on WhatsApp, blogs and social media; Trust alternative sources of information [37, 40, 57]. | Frequency of exposure to positive social media messages [48]. |
| | N.A. | Trust in official sources of information, legacy media (e.g. TV, radio, newspapers, magazines) [37, 44, 57, 75, 79]. Thinks media over report vaccine side effects [41]. |
| **Community and Public Policy** | | |
| **Factors** | **Barriers** | **Facilitators** |
| **Magnitude of community spread** | Number of confirmed or suspected cases in the community [78]. | N.A. |
| **Rate of community vaccination** | Perceived low risk to self when others had been vaccinated [50]. | Acknowledged that vaccines were taken by many of the population [67]. |
| **Choice of vaccination venue** | Mobile vaccination unit [40]. | Local GPs [40]. |
| | Expectations of vaccination venue not met [49]. | N.A. |
| **Cost of vaccine** | High cost/Price [48, 78]. Concerned with cost associated with vaccines [55]. | If vaccines were free [47, 48]. |
| | Those without health insurance [55]. | |
| **Effects of politics** | Political context of the vaccine approval; political skepticism, endorsement by political figures [58, 66, 76, 79, 81, 82]. | Endorsement by public/political figure [58, 82]. |

depending on endorsement of one's political/public figure of choice, this could either be a barrier [58, 66, 76, 79, 81, 82] or facilitator [58, 82] to accepting COVID-19 vaccination. Table 6 presents a full list of other factors (interpersonal, institutional, community, public policy).

## Discussion

COVID-19 vaccine acceptance and uptake are important factors that could help curb the spread and severity of the disease. Our review distils the factors reported around the world

into various levels in the social ecological model revealing the environmental influencing factors at play [19, 83], upon which targeted interventions or policies could be considered. By organizing the factors into barriers and facilitators, the comparison highlighted that the same factor could either promote or deter vaccine acceptance.

In consensus with other reviews, demographic factors such as being female, being younger, having lower education, or having a low income contributed to COVID 19 vaccine hesitancy [11–13, 22, 84]. Similar to the findings by Al-Jayyousi et al [22], our results show that most factors identified are related to intrapersonal level on one's knowledge, attitudes, behavior, self-concepts, and skills. These factors are a culmination of one's experiences and interactions with the environment, which would be difficult to influence quickly. For example, pre-existing factors that impact vaccine acceptance or hesitancy include history and perception of general vaccinations, knowledge of COVID-19 and health-related information, belief in conspiracy theories, as well as personal characteristics and preferences [22].

A major intrapersonal factor that is unique to COVID-19 vaccines is the concern regarding their effectiveness and potential side effects, including their long-term safety, which were also identified in other reviews [11, 12, 22, 76]. These are valid concerns given the accelerated rate of the COVID-19 vaccine development and the lack of long-term safety data [85, 86]. Although many benefitted, and are still benefitting from this unprecedented speed in the history of vaccine development, others cited it as a concern that led to their hesitancy [29, 55, 60, 76, 77]. A recent multi-country survey that tested acceptance of four hypothetical COVID-19 vaccines with varying efficacy and safety profiles found that higher efficacy and lower risks increased the acceptance level among study participants [87]. An interesting point to note from this study was that those believing that new vaccines are riskier than older vaccines were less likely to accept any of the new hypothetical vaccines [87].

Besides intrapersonal factors, common factors found in COVID-19 vaccine acceptance or hesitancy can be broadly summarized into trust in authorities (government/health care including pharmaceutical) and trust in legacy media versus social media [11, 12, 22, 76]. In our review, such factors were categorised under institutional factors. Citing concerns on effectiveness and safety of a vaccine imply some level of doubt in the authorities [88]. Trust and confidence in any authorities stem from historic and existing systems that could not be addressed instantly but would affect people's attitude towards the current recommended vaccines [21, 88, 89]. Understanding weaknesses in the system and investing in better healthcare structure would be longer-range goals that could ultimately address people's trust issues [21]. For more immediate results, communicating consistent information on efficacy and safety of the COVID-19 vaccines could impact people's acceptance of the vaccines [87], which might help combat misinformation in the media and social media.

While influence from family and friends was briefly mentioned in the other reviews, our findings suggested that interpersonal level of influence could play a substantial role in swaying one toward or away from acceptance of the COVID-19 vaccines. One would be more hesitant if they knew someone who had a serious adverse reaction from the vaccine [52] or did not know someone close that was affected by the disease [64], whereas knowing people who were vaccinated, especially one's role model [29, 48, 54], or knowing someone infected with COVID-19 [66], would facilitate vaccine acceptance. However, it could also be the case that individuals tend to socialise with others like them (*i.e.*, someone pro-vaccination would likely socialise with other pro-vaccination individuals). Two studies found that valuing opinions of family or friends who share the same views/beliefs facilitated COVID-19 vaccine acceptance [38, 71]. Hence, the exact nature of the role played by family and friends in vaccination decisions should be explored further in future research so that policies or programs that target those who are hesitant could consider extending beyond that individual.

As demonstrated in the above example, our findings highlighted that even though some factors seemed to present a clearer direction for intervention or policy, others were more context-dependent and not clearly a barrier or facilitator. For instance, having comorbidities or belonging to a high-risk group could be a positive factor that motivates people to be vaccinated [10, 36, 41, 48–50, 53, 54, 61, 62]. Conversely, the same factor presented as a barrier for others who were likely concerned with how the vaccine would impact their medical conditions or health [44, 60]. However, this could be due to an earlier advisory which cautioned against vaccination for those with certain medical conditions, which has since changed [90, 91]. Perceived good health could also pose either as a barrier [9, 52] or a facilitator [61] to one's willingness to be vaccinated. In short, having contextual information on the target population will be crucial to understand factors that pose as barriers and facilitators.

## Implications

Although organizing the factors by respective levels of influence in the social ecological framework provided some distinctions that could inform areas for potential interventions or policies, they are still very much intertwined and pose more questions, such as the exact nature of the role played by family and friends in vaccination decisions. Further research on this relationship might make identifying effective strategies to overcome barriers easier.

Consequently, the need to understand the context, especially of the barriers, should be emphasized, since the same factor could be a facilitator to some while a barrier to others. Structural barriers [92] such as the community and public policy factors affect access and could impact people's acceptance to be vaccinated. Attitudinal barriers [92] such as the factors identified in the intrapersonal, interpersonal, and institutional/organizational levels played a major role in influencing one's acceptance or hesitancy but are more complex to address. Some of these attitudinal barriers could stem from institutional or policy level gaps that would only be known if time and effort were taken to understand them [21]. This could be achieved through engaging targeted subgroups, groups, or communities through partnership [92], such as through dialogues.

In addition, COVID-19 is the first pandemic in the 21st century with unprecedented worldwide aftermath, as well as having on-going impacts on population health and economics [93]. As parts of the world are beginning to recover from the aftermath of the initial COVID-19 infection, COVID-19 has been shown to be a moving target that would continue to influence people's views and acceptance of the vaccines. With scientists warning of more of such infectious diseases in the future [94, 95], it might be important to study the current dynamic response of people's acceptance and hesitancy toward COVID-19 vaccines during different periods of the pandemic. For example, the findings from this review provide some insights into the general population's views on accepting a newly developed vaccine after results of the trials confirmed they were effective and safe. Insights gathered could serve as a guide to future response plans for new infectious disease outbreaks (*e.g.*, by pre-emptively addressing concerns before commencing a nationwide vaccination effort).

Lastly, COVID-19 vaccine rollout prioritized those most at risk, such as healthcare workers and older adults; the latter are the most vulnerable to severe infection and death. As people's needs and situations are heterogeneous, a customized approach to different segments of the population has proven to be both pragmatic and essential. Seeking to understand which factors pose as barriers or facilitators, and for which populations, could help inform context-relevant policies or programs.

## Limitations

Our findings have several limitations. It is possible that some studies have been missed by our search strategy, due to adopting a more general search strategy. Summarizing the

heterogeneous studies was challenging, especially on what vaccine hesitancy entails, which was not explored in depth due to the complexity of the ongoing discussion surrounding the term vaccine hesitancy; thus, we determined it to be beyond the scope of our study. For instance, most studies developed their own surveys to measure vaccine acceptance or hesitancy and the lack of methodological equivalence could cause differences in findings. In addition, we have several inherent limitations when using the social-ecological framework, as classifying factors into the five categories were subjective. Although we minimised the subjectivity through defining sub-categories of each factor in the framework a priori, and performed cross checks to calibrate our understanding and agreement, we acknowledge that subjectivity may not have been completely eliminated.

While there are limitations in using barriers and facilitators, due to the overlapping of factors (as well as not clearly addressing the interrelatedness of those factors) [23], the social- ecological framework has helped to frame those realms, which provided some clarity and insights. The above limitations notwithstanding, our review has identified important barriers and facilitators of vaccine acceptance and hesitancy.

## Conclusion

Our review has identified barriers and facilitators of vaccine acceptance and hesitancy and organised them using the social ecological framework. These factors are context-, population-, and even sub-population-dependent, which could present either as barriers or facilitators. It also shows that factors associated with COVID-19 vaccine acceptance and hesitancy could stem from different levels of influence that are intertwined. Our findings present a general scope of barriers or facilitators that should be considered when developing programs or policies to promote acceptance and uptake of the COVID-19 vaccines, while highlighting the need to also consider the varying contexts experienced by different population.

## Supporting information

**S1 Table. Databases search strategy.**
(DOCX)

**S2 Table. Study characteristics details.**
(DOCX)

**S3 Table. Data extraction sheet.**
(XLSX)

**S4 Table. Excluded studies from full text screening.**
(DOCX)

**S5 Table. Preferred reporting items for systematic reviews and meta-analyses extension for scoping reviews (PRISMA-ScR) checklist.**
(PDF)

## Acknowledgments

We would like to express our sincere thanks to Ms Yasmin Lynda Munro at the Nanyang Technological University, Lee Kong Chian School of Medicine (LKCMedicine) Medical Library, for her advice and assistance on the search strategy and the database searches.

## Author Contributions

**Conceptualization:** Chou Chuen Yu, James Alvin Low, Pradeep Paul George.

**Data curation:** Penny Lun, Jonathan Gao, Bernard Tang.

**Formal analysis:** Penny Lun, Jonathan Gao, Bernard Tang.

**Investigation:** Penny Lun, Jonathan Gao, Bernard Tang.

**Project administration:** Penny Lun.

**Supervision:** Chou Chuen Yu, James Alvin Low, Pradeep Paul George.

**Validation:** Bernard Tang.

**Visualization:** Penny Lun, Jonathan Gao.

**Writing – original draft:** Penny Lun, Jonathan Gao.

**Writing – review & editing:** Penny Lun, Jonathan Gao, Bernard Tang, Chou Chuen Yu, Khalid Abdul Jabbar, James Alvin Low, Pradeep Paul George.

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
