## [Decision Letter · Decision Letter 0]

4 Apr 2022

PONE-D-22-04837A social ecological approach to identify the barriers and facilitators to COVID-19 vaccination acceptance: A scoping reviewPLOS ONE

Dear Dr. George,

Thank you for submitting your manuscript to PLOS ONE. After careful consideration, we feel that it has merit but does not fully meet PLOS ONE’s publication criteria as it currently stands. Therefore, we invite you to submit a revised version of the manuscript that addresses the points raised during the review process.

We look forward to receiving your revised manuscript.

Kind regards,

Harapan Harapan, MD, PhD

Academic Editor

PLOS ONE

Journal Requirements:

Additional Editor Comments:

1. Abstract and Introduction: Please define the COVID-19 first.

2. Method: Please be consistent in this part: "COVID-19 or nCoV* or 2019nCoV or 19nCoV or COVID19* or COVID or SARS-COV-2 or SARSCOV-2 or SARSCOV2 or "Severe Acute Respiratory Syndrome Coronavirus 2" [20] or "Severe Acute Respiratory Syndrome Corona Virus 2" and Vaccination Refusal/ or Anti-Vaccination Movement/ or Mass Vaccination/ or Vaccination Coverage/ (hesitan* or acceptance or preference or rejection or anti-vaccin* or attitude? or barrier? or facilitator? or intent*)."

3. Discussion: Please include the finding of these references briefly and cite: https://doi.org/10.52225/narra.v1i3.55 AND https://doi.org/10.52225/narra.v1i3.57

Reviewers' comments:

Reviewer's Responses to Questions

**Comments to the Author**

1. Is the manuscript technically sound, and do the data support the conclusions?

Reviewer #1: Yes

Reviewer #2: Partly

Reviewer #3: Yes

2. Has the statistical analysis been performed appropriately and rigorously? 

Reviewer #1: N/A

Reviewer #2: N/A

Reviewer #3: N/A

3. Have the authors made all data underlying the findings in their manuscript fully available?

Reviewer #1: No

Reviewer #2: Yes

Reviewer #3: No

4. Is the manuscript presented in an intelligible fashion and written in standard English?

Reviewer #1: Yes

Reviewer #2: Yes

Reviewer #3: Yes

5. Review Comments to the Author

Reviewer #1: Thanks a lot for the opportunity to review this interesting and timely manuscript

In the current scoping review, the authors aimed to identify the factors affecting COVID19 vaccine hesitancy, using the social ecological model. This model has wide applications including those in the field of public health and is considered as a suitable framework for disease prevention.

Overall, the manuscript is well-written, the background can benefit from further elaboration on COVID-19 vaccine acceptance and hesitancy worldwide (please see comments below). The results were clear, as well as the implications of this review. The limitations were addressed clearly, and the conclusions were supported by the results.

I have the following minor points that hopefully can help the authors to improve the final manuscript:

1. Please try to rephrase the following statement in the Abstract: “Vaccine hesitancy is at its highest following the introduction of any new vaccine” since it was found the following paper with the same wording: https://www.ncbi.nlm.nih.gov/pmc/articles/PMC8030992/

2. In the Abstract please use “younger age” instead of “lower age”

3. In the Introduction, the paragraph starting with “Like most countries, one of the first segments of population targeted for COVID-19 vaccination in Singapore…” seems out of context. Please revise by providing a general overview of COVID-19 vaccine acceptance and hesitancy citing the following relevant references:

A. https://www.ncbi.nlm.nih.gov/pmc/articles/PMC8760993/

B. https://www.ncbi.nlm.nih.gov/pmc/articles/PMC7920465/

4. I was not able to access Supporting information (S1 – S5 Tables). Hopefully you can provide this valuable data.

5. In the results, “Majority of the studies were cross-sectional studies (n=32, 62%)” I think that 32/51 will be 63% rather than 62%. Please revise

6. In the results “Most of the studies had sample sizes of over 1000 (n=32, 63%), and included participants from Europe (n=15, 29%) and North America (n=15, 29%), followed by Asia (n=19, 18%), the Middle East (n=5, 10%)” The sum of these studies exceeds 51. Please revise

7. In Table 2, please correct the percentage of cross-sectional studies (63% instead of 62%). Actually, the majority of percentages need checking since for the same number (5) two different percentages were reported 10 vs 12%. Please revise.

8.

Reviewer #2: - Summary

o The article presents a scoping review of factors influencing COVID-19 vaccine hesitancy and acceptance using a social ecological framework. The topic is very timely, and the results are interesting to read and could have important implications for policy and practice if the points below are addressed.

o The term ‘hesitancy’ is used to imply barriers and acceptance to imply facilitators, but this is not how these terms are used in literature. Also, viewing these factors as either barriers or facilitators downplays the complex interplay of factors influencing vaccine decision-making, particularly in the context of ‘new’ vaccines like the COVID-19 vaccines with widespread media attention and dis- and misinformation, and so on.

o There is a lack of critical engagement with current literature. The term ‘vaccine hesitancy’ is used in this article yet prominent researchers in this space (e.g., Eve Dube, Julie Leask) are not referenced. Moreover, there is much debate in literature about the term ‘vaccine hesitancy’ and this requires a reflection in the review of how the term is used and the implications of this. The authors may wish to read these articles for example:

Attwell et al., 2022 https://www.nature.com/articles/d41586-022-00495-8

Bedford et al., 2018 https://pubmed.ncbi.nlm.nih.gov/28830694/

o The authors briefly note that a focus on factors for different age groups (older and younger adults); however, this is not woven in throughout the review. If this is indeed the focus of the review, then the authors are recommended to focus on older adults (and this appears appropriate given the authors’ expertise) from the outset. That being said, considering only one article from their search focused on older adults, this may not be possible.

o As much of the information presented in the review is known, there is an area of opportunity to clarify how this review advances our knowledge and implications for policy, practice and future research.

- Abstract

o Background

Please change “significant” to “substantial”

The pandemic has not only impacted healthcare systems. Please consider noting that the pandemic has had widespread social, economic and health impacts.

“One of the biggest obstacles to implementing successful vaccination programmes is vaccine hesitancy. Vaccine hesitancy is at its highest following the introduction of any new vaccine”

• This sentence really depends on how vaccine hesitancy is being used. As above, there is much debate about how this term overlooks systemic and logistical aspects that present important barriers to vaccination. Also, most literature discusses how vaccine hesitancy is influenced by concerns of vaccine safety.

o Methods

Please include brief details about the inclusion/exclusion criteria.

o Results

Please include some specific findings related to the interpersonal, institution, community and public policy levels.

o Conclusion

These comments can be improved to better highlight how this review advances our knowledge and implications to policy and practice.

- Introduction

o “COVID-19 has caused significant impact on the health of the world’s population, with over 356 million confirmed cases and 5.6 million deaths worldwide as of 27th January 2022 [1].”

Please update figures.

o “to stem the flow of infections”

Perhaps change to “reduce the spread”

o “safety/effectiveness”

Safety and effectiveness refers to two separate points, please re-consider using a hyphen in this context.

o “one of the first segments of the population”

o “Despite the current high vaccination rate,”

Is this referring to the national rates in Singapore? If yes, please be specific.

This paragraph should be expanded upon with more global literature, particularly if the focus will be on older adults.

o End of p.4 –

As above, this focus on older adults appears out of place unless the focus of the review will be on different age groups. However, if the review will focus on wider population groups, then referencing additional literature is warranted. It is noted that factors will differ according to context and population sub-group (e.g., parental vaccine hesitancy vs adult vaccine hesitancy).

o Considering the focus is on vaccine hesitancy, there must be more critical engagement with the literature and discussing factors already known to influence vaccine uptake. While we cannot necessarily infer that these factors will also be applicable to the COVID-19 vaccines, it is warranted to note factors that are already reported in literature regarding routine vaccines.

o “Hence, to better understand the factors that determine one’s willingness or intentions to accept the vaccines”

Suggest specifying that the authors are referring to COVID-19 vaccines.

Please clarify who “one’s willingness or intention to accept the vaccine” is referring to. For instance, is it older adults or another group or in general?

o The end of the Introduction would benefit from noting what other reviews have already been conducted in this space to support justification for conducting this review.

- Methods

o The authors have used an appropriate framework for conducting a scoping review (i.e., Arksey & O’Malley’s framework).

o The authors are encouraged to mention the purposes of doing a scoping review (as per Arksey & O’Malley) and justify why a scoping review was chosen as opposed to other review types (e.g., rapid review, systematic review).

o “with advancements proposed by…”

What is this specifically referring to?

o “Quality assessments were not performed to preserve the entirety of the studies found [18].”

What does “preserve the entirety of the studies found”? Kindly clarify.

As assessment for quality is an aspect that can set a scoping review apart from a systematic review, it is important to clarify this decision.

o The research question

“This scoping review aims to study the barriers and facilitators that influence COVID-19 vaccine worldwide and to ascertain if these factors differ between older (65 and above) and younger adults.”

• As above, this focus on age is not appropriately woven in at the outset. Thus, the authors are recommended to include supporting justification and references in the Abstract and Introduction.

“This review will also inform the design of a study to explore older adults’ attitudes towards COVID-19 vaccination”

• What is this specifically referring to? Is this a follow-up study that the authors are involved in? Perhaps clarification could be included please.

o Identifying relevant studies

The search strategy is sound.

The end date was April 14, 2021.

• Given how much the literature space is evolving, the authors are recommended to run an updated search to include the most recent literature.

o Study selection

It appears that the time frame was Sept 2020 to April 2021; the time frame could be better clarified. As above, the authors are recommended to re-run their search for more recent literature.

The authors are encouraged to clarify if all study designs were included?

- Results

o Barriers and facilitators using socio-ecological framework

Socio-demographic determinants

• Did any studies report deprivation level or socio-economic status being a factor?

Interpersonal factors

• Did any references discuss the influence of friends and family? And also, any references discuss the influence of health providers’ recommendations?

Institutional/organisational factors

• Was there any studies that discussed access issues?

Community factors and public policy

• Did any studies discuss the influence of mandates and any other public health policies?

- Discussion

o As above, the Discussion section would benefit from a more critical engagement with literature.

The review mentions how “most of these studies assessed one’s willingness or intentions to take the COVID-19 vaccines when available, using Likert-type scales.” As there are many tools used to measure vaccine hesitancy and acceptance (and confidence, etc.), in addition to how these terms are used differently in literature, a discussion about what tools the included studies used and how these may or may not be able to be compared and why is warranted.

o As above, the implications and conclusions of the study can be strengthened.

Reviewer #3: One of the objectives of scoping review is to find the research gaps where more targeted work is needed to accomplish the programmatic goals. Under the discussion and implication section this could have been done with more clarity. Under policy, the vaccine availability issues have not been explored adequately. The cost of vaccine along with private sector engagement and people's willingness to pay are also not discussed. The older adults' physical access challenges along with its implications for non-covid vaccination could have been discussed further.

6. PLOS authors have the option to publish the peer review history of their article (what does this mean?). If published, this will include your full peer review and any attached files.

Reviewer #1: No

Reviewer #2: No

Reviewer #3: **Yes: **Arindam Ray

---

## [Author Response · Author response to Decision Letter 0]

7 Jul 2022

Dear Dr Harapan,

We would like to express our sincere thanks to you and the reviewers for the invaluable comments that have helped to improve our manuscript. We have addressed the comments and made the necessary changes as much as possible in the manuscript. Please note that we have also made additional edits and changes (highlighted in the marked copy) where needed. In addition, we are sharing all the dataset for this study through the current updated supplementary materials (S1-S4) to adhere to PLOS Data Policy. 

The point-to-point reply to you and the reviewers can be found below.

EDITOR

1. Abstract and Introduction: Please define the COVID-19 first.

Reply: COVID-19 has now been defined in both the abstract and the introduction section.

Amendments: Pg. 2 first line and pg. 4 first line

2. Method: Please be consistent in this part: "COVID-19 or nCoV* or 2019nCoV or 19nCoV or COVID19* or COVID or SARS-COV-2 or SARSCOV-2 or SARSCOV2 or "Severe Acute Respiratory Syndrome Coronavirus 2" [20] or "Severe Acute Respiratory Syndrome Corona Virus 2" and Vaccination Refusal/ or Anti-Vaccination Movement/ or Mass Vaccination/ or Vaccination Coverage/ (hesitan* or acceptance or preference or rejection or anti-vaccin* or attitude? or barrier? or facilitator? or intent*)."

Reply: We have reviewed and edited this section to clarify the key terms that guided the search strategy for the databases. The Covid-19 search terms were derived from CADTH COVID-19 search strings (see reference 20). In addition, we have provided the search strategy that was adopted in the databases in the current updated S1 Table.

Amendments: Paragraph in “Identifying relevant studies” under Methods section.

3. Discussion: Please include the finding of these references briefly and cite: https://imsva91-ctp.trendmicro.com:443/wis/clicktime/v1/query?url=https%3a%2f%2fdoi.org%2f10.52225%2fnarra.v1i3.55&umid=3EB52A1C-DCA7-1205-89F5-1758E27F334C&auth=6e3fe59570831a389716849e93b5d483c90c3fe4-39466bce0dd84bcd2dca354b828e4cc6a2a74c6f AND https://imsva91-ctp.trendmicro.com:443/wis/clicktime/v1/query?url=https%3a%2f%2fdoi.org%2f10.52225%2fnarra.v1i3.57&umid=3EB52A1C-DCA7-1205-89F5-1758E27F334C&auth=6e3fe59570831a389716849e93b5d483c90c3fe4-7966a15ce4a4590c70705dbacbd971aa047410a8

Reply: Thank you. We have included the article by Hassan et al. (2021) as a citation and have discussed the article by Rosiello et al. (2021) to help strengthen the discussion section.

Amendments: 2nd to 4th para. in discussion section.

REVIEWER #1:

Thanks a lot for the opportunity to review this interesting and timely manuscript

In the current scoping review, the authors aimed to identify the factors affecting COVID19 vaccine hesitancy, using the social ecological model. This model has wide applications including those in the field of public health and is considered as a suitable framework for disease prevention. Overall, the manuscript is well-written, the background can benefit from further elaboration on COVID-19 vaccine acceptance and hesitancy worldwide (please see comments below). The results were clear, as well as the implications of this review. The limitations were addressed clearly, and the conclusions were supported by the results.

I have the following minor points that hopefully can help the authors to improve the final manuscript:

Reply: We are grateful for your encouraging comments. Thank you.

1. Please try to rephrase the following statement in the Abstract: “Vaccine hesitancy is at its highest following the introduction of any new vaccine” since it was found the following paper with the same wording: https://imsva91-ctp.trendmicro.com:443/wis/clicktime/v1/query?url=https%3a%2f%2fwww.ncbi.nlm.nih.gov%2fpmc%2farticles%2fPMC8030992%2f&umid=3EB52A1C-DCA7-1205-89F5-1758E27F334C&auth=6e3fe59570831a389716849e93b5d483c90c3fe4-58baa76331b72c6b9db3da2f39d4de81497f506d

Reply: Thank you for highlighting this oversight. We decided to remove this statement because it did not contribute to abstract in a significant way.

2. In the Abstract please use “younger age” instead of “lower age”

Reply: We have amended this accordingly.

Amendment: Abstract “Results” section.

3. In the Introduction, the paragraph starting with “Like most countries, one of the first segments of population targeted for COVID-19 vaccination in Singapore…” seems out of context. Please revise by providing a general overview of COVID-19 vaccine acceptance and hesitancy citing the following relevant references:

https://imsva91-ctp.trendmicro.com:443/wis/clicktime/v1/query?url=https%3a%2f%2fwww.ncbi.nlm.nih.gov%2fpmc%2farticles%2fPMC8760993%2f&umid=3EB52A1C-DCA7-1205-89F5-1758E27F334C&auth=6e3fe59570831a389716849e93b5d483c90c3fe4-e54f517ecaa562ec9acf88d9bbe6b63ce85243d5

B. https://imsva91-ctp.trendmicro.com:443/wis/clicktime/v1/query?url=https%3a%2f%2fwww.ncbi.nlm.nih.gov%2fpmc%2farticles%2fPMC7920465%2f&umid=3EB52A1C-DCA7-1205-89F5-1758E27F334C&auth=6e3fe59570831a389716849e93b5d483c90c3fe4-d6b8462702582c3fe0f267439280afc85b9517ea

Reply: Thank you for suggesting these articles. As part of the revision of the manuscript, this paragraph has been revised to be more general with inclusion of these recommended systematic reviews.

Amendments: Para. 2 in the introduction section.

4. I was not able to access Supporting information (S1 – S5 Tables). Hopefully you can provide this valuable data.

Reply:

We’re sorry that you could not access the supplementary materials, as they were uploaded during the submission. As part of the revision, we have moved the former S3-S5 tables that contained the result to the main manuscript for easier access to the information; they have been renamed Tables 4 to 6 respectively. In addition, the revised S1 Table contained our search strategy in all the databases now and S3 and S4 Tables are new supplementary materials that contained data to our study (data extraction sheet and excluded references after the full text screening process. Only S2 Table remained unchanged from the initial submission.

5. In the results, “Majority of the studies were cross-sectional studies (n=32, 62%)” I think that 32/51 will be 63% rather than 62%. Please revise

6. In the results “Most of the studies had sample sizes of over 1000 (n=32, 63%), and included participants from Europe (n=15, 29%) and North America (n=15, 29%), followed by Asia (n=19, 18%), the Middle East (n=5, 10%)” The sum of these studies exceeds 51. Please revise

7. In Table 2, please correct the percentage of cross-sectional studies (63% instead of 62%). Actually, the majority of percentages need checking since for the same number (5) two different percentages were reported 10 vs 12%. Please revise.

Reply: Thank you for highlighting these points. We have amended the above points respectively; the discrepancies are due to uneven rounding off of the decimal points.

Amendments: Table 2

REVIEWER #2:

o The article presents a scoping review of factors influencing COVID-19 vaccine hesitancy and acceptance using a social ecological framework. The topic is very timely, and the results are interesting to read and could have important implications for policy and practice if the points below are addressed.

Reply: We are grateful for your encouraging comments. Thank you.

o The term ‘hesitancy’ is used to imply barriers and acceptance to imply facilitators, but this is not how these terms are used in literature. Also, viewing these factors as either barriers or facilitators downplays the complex interplay of factors influencing vaccine decision-making, particularly in the context of ‘new’ vaccines like the COVID-19 vaccines with widespread media attention and dis- and misinformation, and so on

Reply: We agree that the studies on attitudes towards vaccine acceptance/hesitancy are complex and complicated with varying definitions. We have added this point on the complexity in the introduction, as well as acknowledged the limitations of using barriers and facilitators in the “Limitations” section. In addition, we have amended to define barriers and facilitators in our study and referenced 2 studies that had approached vaccine acceptance using the barriers and facilitators concept.

Amendments: Introduction para. 3 (Pg 5), “Charting the data” para. 1 under the “Methods” section (pg 8), “Limitations” section para. 1 (pg 29).

o There is a lack of critical engagement with current literature. The term ‘vaccine hesitancy’ is used in this article yet prominent researchers in this space (e.g., Eve Dube, Julie Leask) are not referenced. Moreover, there is much debate in literature about the term ‘vaccine hesitancy’ and this requires a reflection in the review of how the term is used and the implications of this. The authors may wish to read these articles for example:

Attwell et al., 2022 https://imsva91-ctp.trendmicro.com:443/wis/clicktime/v1/query?url=https%3a%2f%2fwww.nature.com%2farticles%2fd41586%2d022%2d00495%2d8&umid=3EB52A1C-DCA7-1205-89F5-1758E27F334C&auth=6e3fe59570831a389716849e93b5d483c90c3fe4-6aedcfddab62fad68ffaf29236658acb1b008e8d

Bedford et al., 2018 https://imsva91-ctp.trendmicro.com:443/wis/clicktime/v1/query?url=https%3a%2f%2fpubmed.ncbi.nlm.nih.gov%2f28830694%2f&umid=3EB52A1C-DCA7-1205-89F5-1758E27F334C&auth=6e3fe59570831a389716849e93b5d483c90c3fe4-3f51f62b722a9ffe49e6f1264ff823edcf9b531e

Reply:

Thank you for bringing our attention to these relevant articles that helped us to reflect on the debate on vaccine hesitancy. Again, we agree that vaccine hesitancy is a very complex topic and the authors of both articles have indicated the need to distill the term “hesitancy” further which could include wider system-related factors. We believe that the social-ecological model has helped teased out such factors under interpersonal, institutional, community and public policy level related barriers that both Attwell et al, 2022 and Bedford et al., 2018 have highlighted. As such, we have integrated both articles into various sections in the manuscript to better reflect the complexity and its implications more, although also acknowledging that there are limitations on not having gone in-depth due to the scope of our study. In addition, we have also clarified on what was included in the hesitancy rate that was presented in the “Results-Acceptance/Hesitancy Rate” section.

Amendment: Introduction section para. 3 (pg 5), Discussion section para. 4 (pg 26) Implication section para. 2 (pg 21), “Limitations” section para. 1 (pg 27-28). “Acceptance/Hesitancy Rate” in “Results” section para. 1 (pg 10-11). 

o The authors briefly note that a focus on factors for different age groups (older and younger adults); however, this is not woven in throughout the review. If this is indeed the focus of the review, then the authors are recommended to focus on older adults (and this appears appropriate given the authors’ expertise) from the outset. That being said, considering only one article from their search focused on older adults, this may not be possible.

Reply:

Thank you for helping us to further difine our direction. We were initially interested to focus on older adults, but as we reviewed the articles, this was not possible. Hence we ended up focusing on adults in the general public, which also included older adult participants in the surveys. We have revised the narrative of the manuscript to reflect this focus on general adult population better. As a results previous focus on older adults was removed and the manuscript has been aligned to focusing on adults (including older adults) overall. The research question has been amended to correctly reflect this alignment, as well as the inclusion criteria. 

Amendment: “The research question” under “Methods” section. Inclusion criteria Table 1 from “adults and older adults” to “adults (15 and above)”, as several surveys included participants from 15 and 16 onwards. 

o As much of the information presented in the review is known, there is an area of opportunity to clarify how this review advances our knowledge and implications for policy, practice and future research.

Reply: We concede that in the original manuscript, we had missed the opportunity to clarify on the advancement of the knowledge from our review and its implications clearly. As such, we have revised and added to the introduction and discussion section to highlight the contributions of our review, in terms of the scope and focus, in order to take time sensitivity of barriers and facilitators into consideration. 

Amendment: Last 2 para. in the “Introduction” section (pg), first para. in Discussion section (pg 25), Limitations section para. 2.

Abstract

o Background

Please change “significant” to “substantial”

The pandemic has not only impacted healthcare systems. Please consider noting that the pandemic has had widespread social, economic and health impacts.

Reply: We have amended both points in this section according to your suggestions.

Amendment: Abstract “Background” (pg 2)

“One of the biggest obstacles to implementing successful vaccination programmes is vaccine hesitancy. Vaccine hesitancy is at its highest following the introduction of any new vaccine”

• This sentence really depends on how vaccine hesitancy is being used. As above, there is much debate about how this term overlooks systemic and logistical aspects that present important barriers to vaccination. Also, most literature discusses how vaccine hesitancy is influenced by concerns of vaccine safety.

Reply: We have revised the first statement to indicate that vaccine hesitancy stems from concerns with the effectiveness and safety. The second sentence has been removed as it was flagged by reviewer #1 and did not contribute to the abstract in a significant way. 

Amendment: Abstract “Background” (pg. 2). 

o Methods Abstract “Background”

Please include brief details about the inclusion/exclusion criteria.

Reply: We have included a statement on the selection criteria framework of PICo, but were unable to expand further due to word limitations.

Amendment: Abstract “Methods” section (pg. 2)

o Results

Please include some specific findings related to the interpersonal, institution, community and public policy levels.

Reply: Due to the word count limitation, we could only highlight the specifics under intrapersonal level, which contained the major part of the results. 

o Conclusion

These comments can be improved to better highlight how this review advances our knowledge and implications to policy and practice.

Reply: We agree that the conclusion could be improved to highlight contributions of this review and have revised the conclusion to highlight the implication on policy.

Amendment: Abstract “Conclusion” section. 

Introduction:

o “COVID-19 has caused significant impact on the health of the world’s population, with over 356 million confirmed cases and 5.6 million deaths worldwide as of 27th January 2022 [1].”

Please update figures.

o “to stem the flow of infections”

Perhaps change to “reduce the spread”

Reply: We have amended the above as suggested.

Amendment: Introduction section para. 1 (pg. 4).

o “safety/effectiveness”

Safety and effectiveness refers to two separate points, please re-consider using a hyphen in this context.

Reply: This statement has been revised due to revision made in the introduction section.

Amendment: Introduction para. 2 (pg. 4).

o “one of the first segments of the population”

o “Despite the current high vaccination rate,”

Is this referring to the national rates in Singapore? If yes, please be specific.

This paragraph should be expanded upon with more global literature, particularly if the focus will be on older adults.

Reply: We agree that this paragraph should be more global and not focusing on Singapore. It has been revised accordingly with additional general literature to present a more global view.

Amendment: Introduction section para. 2 & 3 (pg. 4-5)

As above, this focus on older adults appears out of place unless the focus of the review will be on different age groups. However, if the review will focus on wider population groups, then referencing additional literature is warranted. It is noted that factors will differ according to context and population sub-group (e.g., parental vaccine hesitancy vs adult vaccine hesitancy).

Reply: As indicated under the summary comments, we have removed the focus on older adults to focus on overall adults including older adults. In addition, we agree that the results might differ among subgroup and hence had only included studies with participants from the general public from the beginning.

o Considering the focus is on vaccine hesitancy, there must be more critical engagement with the literature and discussing factors already known to influence vaccine uptake. While we cannot necessarily infer that these factors will also be applicable to the COVID-19 vaccines, it is warranted to note factors that are already reported in literature regarding routine vaccines.

Reply: We have revised the introduction to include more literature on COVID-19 vaccine acceptances and hesitancy as well as including a review on general vaccine confidence and barriers to uptake. As COVID-19 vaccines entailed certain unique factors different from other vaccines, we did not focus on literature that explored general vaccine uptake. 

Amendment: Introduction para. 2 & 3 (pg 4-5).

o “Hence, to better understand the factors that determine one’s willingness or intentions to accept the vaccines”

Suggest specifying that the authors are referring to COVID-19 vaccines.

Please clarify who “one’s willingness or intention to accept the vaccine” is referring to. For instance, is it older adults or another group or in general?

Reply: We have edited this sentence to specify COVID-19 vaccines and clarify the focus on general public adult population.

Amendment: Introduction para. 4 (pg. 5).

o The end of the Introduction would benefit from noting what other reviews have already been conducted in this space to support justification for conducting this review.

Reply: We have addressed this point in the last paragraph by referencing other scoping reviews and justification for this review.

Amendment: Introduction last para. (pg. 5).

Methods

o The authors have used an appropriate framework for conducting a scoping review (i.e., Arksey & O’Malley’s framework).

o The authors are encouraged to mention the purposes of doing a scoping review (as per Arksey & O’Malley) and justify why a scoping review was chosen as opposed to other review types (e.g., rapid review, systematic review).

o “with advancements proposed by…”

What is this specifically referring to?

Reply: We have included a statement to explain the use of scoping review, which also tie in with the next point on quality assessment. The advancement refers to improvements in the framework which has been added. One example of this improvement was provided in the “study selection” under Methods.

Amendment: Methods section para. 1 (pg. 5-6).

o “Quality assessments were not performed to preserve the entirety of the studies found [18].”

What does “preserve the entirety of the studies found”? Kindly clarify.

As assessment for quality is an aspect that can set a scoping review apart from a systematic review, it is important to clarify this decision.

Reply: Please refer to the point above, the statement has been amended to make it clearer.

Amendment: Methods section para. 1 (pg. 5-6).

o The research question

“This scoping review aims to study the barriers and facilitators that influence COVID-19 vaccine worldwide and to ascertain if these factors differ between older (65 and above) and younger adults.”

• As above, this focus on age is not appropriately woven in at the outset. Thus, the authors are recommended to include supporting justification and references in the Abstract and Introduction.

Reply: As indicated under the summary and introduction sections comments, we have removed the focus on older adults. The manuscript is now aligned to focusing on the general adults.

“This review will also inform the design of a study to explore older adults’ attitudes towards COVID-19 vaccination”

• What is this specifically referring to? Is this a follow-up study that the authors are involved in? Perhaps clarification could be included please.

Reply: Although the intention initially was to use the information for a follow up study in Singapore, this has since been revised as we did not find enough information on older adults from this scoping review. We have hence removed this statement.

o Identifying relevant studies

The search strategy is sound.

The end date was April 14, 2021.

• Given how much the literature space is evolving, the authors are recommended to run an updated search to include the most recent literature.

Reply: While we agree that the literature is evolving in the topic, the suggestion to conduct an update search was not considered, because of the following reasons:

1) The included studies focused on the initial few months following publication of clinical trial results and identification of the first COVID-19 variant. As the pandemic situation has evolved and will continue to evolve, studies on COVID-19 vaccine acceptance and hesitancy will likely continue, it is hence difficult to chase a moving target. In addition, as the barriers and facilitators are time sensitive and changing, including more recent studies might change the current narrative, which we would like to preserve, for possible future comparison.

2) Our search was comprehensive including several databases, updating them would also require additional time/resources.

3) An update study would be considered at a later date, as the dynamic situation rendered need for repeat studies. 

o Study selection

It appears that the time frame was Sept 2020 to April 2021; the time frame could be better clarified. As above, the authors are recommended to re-run their search for more recent literature.

The authors are encouraged to clarify if all study designs were included?

Reply: Kindly refer to our reply in the previous comment regarding your recommendation to re-run the search. We have edited this section on the time frame and Table 1 to clarify our rationale for the study selection, which was based on data collection date and not publication date. In addition, we have included “all types” in the study inclusion criteria in Table 1.

Amendment: “Study selection” under Methods section para. 1 and Table 1 (pg. 6-7).

Results

o Barriers and facilitators using socio-ecological framework

Socio-demographic determinants

• Did any studies report deprivation level or socio-economic status being a factor?

Reply: Higher socio-economic status was a facilitator identified in only one study, while the others focus on education level and income level as separate factors. These are documented in Table 4 in this revised version, but not described in the text. 

Interpersonal factors

• Did any references discuss the influence of friends and family? And also, any references discuss the influence of health providers’ recommendations?

Reply: Yes. Some aspects were described in text under “Interpersonal factors” section, while the full list is documented in Table 6 in the revised manuscript. In addition, we have added a line in this section to emphasize the influence from family and friends. In addition, we have also added a paragraph in the discussion section on the influence from family and friends.

On the point of influence of healthcare providers, we have added a line to institutional factors (which we have categorized under this) to explain that this could be a barrier or facilitator.

Amendment: Results section under “Interpersonal factors” end of the para. (pg 21) and under “Institutional/organizational factors” para. 2 (pg 21) respectively, and in Table 6 (pg. 23-25). Discussion section para. 5 (pg 27).

Institutional/organisational factors

• Was there any studies that discussed access issues?

Reply: Yes. Accessibility is documented in Table 5 in this revised version under “Intrapersonal factors’ and we have added a line in the Results section to highlight this point. We have accessibility under Intrapersonal instead of Institutional/Organisational due to the way accessibility was framed in these studies. However, the concern with cost of the vaccines was categorised under public policy section and we have added a line to highlight cost in this section. 

Amendment: Results section under “Intrapersonal section” para. 3 (pg. 17) and Table 5 (pg. 18-20). Results section under “Community factor and public policy” para. 1 (pg. 22) and Table 6 (pg. 23-25)

Community factors and public policy

• Did any studies discuss the influence of mandates and any other public health policies?

Reply: Only one study reported that those believing in mandatory vaccinations were more likely to accept the vaccine. This is documented in Table 5 under intrapersonal factors due to the way it was framed. We have added a line in this section to highlight this point. With regards to other public health policies such as preventive measures, those factors were reported more from a personal practice/preference angle and were hence captured under intrapersonal section under “Personal characteristics/preferences” in the same Table 5. 

Amendment: Results section under “intrapersonal factors” para. 2 (pg. 16) and Table 5 (pg. 18-20).

Discussion

o As above, the Discussion section would benefit from a more critical engagement with literature.

The review mentions how “most of these studies assessed one’s willingness or intentions to take the COVID-19 vaccines when available, using Likert-type scales.” As there are many tools used to measure vaccine hesitancy and acceptance (and confidence, etc.), in addition to how these terms are used differently in literature, a discussion about what tools the included studies used and how these may or may not be able to be compared and why is warranted.

Reply: We agree that our discussion section could be strengthened with better engagement with the literature and have improved this section overall. However, as most of the studies did not use a standardized scale, we did not explore nor go in depth on the use of tools. As such, we have added a line in the results section “Acceptance/Hesitancy rate” to clarify that most studies did not use validated scales for the measurement on vaccine hesitancy and have reported this as a limitation.

Amendment: Results section under “Acceptance/Hesitancy Rate” para. 1 (pg. 11) and Limitations section para. 1 (pg 29). 

o As above, the implications and conclusions of the study can be strengthened.

Reply: We agree that our implications and conclusion section could be strengthened. Hence, along the revision in the discussion section, we have made significant amendment to the implication section and amended conclusion to reflect these changes. 

Amendment: Implications, limitations and conclusion section (pg. 25-30). 

REVIEWER #3:

One of the objectives of scoping review is to find the research gaps where more targeted work is needed to accomplish the programmatic goals. Under the discussion and implication section this could have been done with more clarity.

Reply: Thank you for your comment. We agree that our manuscript could be improved to indicate clearer implications. As such, we have made significant changes to the discussion and implication sections to highlight the factors clearer. 

Amendment: Implications, limitations and conclusion section (pg. 25-30). 

Under policy, the vaccine availability issues have not been explored adequately. The cost of vaccine along with private sector engagement and people's willingness to pay are also not discussed. The older adults' physical access challenges along with its implications for non-COVID-19 vaccination could have been discussed further.

Reply: Thank you for this feedback. The cost of the vaccines was only mentioned in a few studies and were documented in Table 6. We have now added a sentence in the results section under Public Policy to present this. However, due to the limitation on the information provided within the studies, we did not expand the information beyond this. We did not include studies that focused solely on the willingness to pay and hence, do not have information on this. With regards to the point on older adults, we have now aligned our focus to adults for the review and hence, our discussion is focused on the general results we have found.

The manuscript has been revised and checked through for readability, with a marked copy highlighting where the changes were made. Thank you.

 Regards,

Regards,

Pradeep Paul George

---

## [Decision Letter · Decision Letter 1]

25 Jul 2022

A social ecological approach to identify the barriers and facilitators to COVID-19 vaccination acceptance: A scoping review

PONE-D-22-04837R1

Dear Dr. George,

We’re pleased to inform you that your manuscript has been judged scientifically suitable for publication and will be formally accepted for publication once it meets all outstanding technical requirements.

Kind regards,

Harapan Harapan, MD, PhD

Academic Editor

PLOS ONE

Additional Editor Comments (optional):

Reviewers' comments:

Reviewer's Responses to Questions

**Comments to the Author**

1. If the authors have adequately addressed your comments raised in a previous round of review and you feel that this manuscript is now acceptable for publication, you may indicate that here to bypass the “Comments to the Author” section, enter your conflict of interest statement in the “Confidential to Editor” section, and submit your "Accept" recommendation.

Reviewer #1: All comments have been addressed

Reviewer #2: All comments have been addressed

2. Is the manuscript technically sound, and do the data support the conclusions?

Reviewer #1: Yes

Reviewer #2: Yes

3. Has the statistical analysis been performed appropriately and rigorously? 

Reviewer #1: Yes

Reviewer #2: N/A

4. Have the authors made all data underlying the findings in their manuscript fully available?

Reviewer #1: Yes

Reviewer #2: Yes

5. Is the manuscript presented in an intelligible fashion and written in standard English?

Reviewer #1: Yes

Reviewer #2: Yes

6. Review Comments to the Author

Reviewer #1: Thanks for addressing all the previous comments properly. I have no further comments or remarks to this manuscript

Reviewer #2: Thank you for addressing the comments, the revised manuscript reads well and will make an important contribution to knowledge as we continue to deal with the COVID-19 pandemic. One final comment is that the authors may wish to contextualise the last sentence of the Abstract to focus on COVID-19 or new vaccines.

7. PLOS authors have the option to publish the peer review history of their article (what does this mean?). If published, this will include your full peer review and any attached files.

Reviewer #1: No

Reviewer #2: No

---

## [Editor Report · Acceptance letter]

23 Sep 2022

PONE-D-22-04837R1 

A social ecological approach to identify the barriers and facilitators to COVID-19 vaccination acceptance: A scoping review 

Dear Dr. Lun:

I'm pleased to inform you that your manuscript has been deemed suitable for publication in PLOS ONE. Congratulations! Your manuscript is now with our production department. 

Kind regards, 

on behalf of

Dr. Harapan Harapan 

Academic Editor

PLOS ONE